# Quantity and Quality Changes in Sugar Beet (*Beta vulgaris* Provar. Altissima Doel) Induced by Different Sources of Biostimulants

**DOI:** 10.3390/plants11172222

**Published:** 2022-08-27

**Authors:** Marek Rašovský, Vladimír Pačuta, Ladislav Ducsay, Dominika Lenická

**Affiliations:** Institute of Agronomic Sciences, Faculty of Agrobiology and Food Resources, Slovak University of Agriculture in Nitra, Tr. A. Hlinku 2, 949 76 Nitra, Slovakia

**Keywords:** sugar beet, biostimulants, variety, yield, quality, climate changes

## Abstract

The application of biostimulants in agriculture is considered an economically and ecologically acceptable and, above all, a sustainable method of cultivation of field crops. This study aimed to investigate the impact of biostimulating agents on the production and growth parameters of the sugar beet. In 2018 and 2019, an experiment was conducted in which the effect of four types of treatment (B0–B3) on two varieties of sugar beet (Alpaca, Gorila) was observed. The results show that the beets treated with treatment type B3 (combination of humic acids, essential amino acids, biopolymers, and soil bacteria) had the significantly highest yield of roots compared with the control type. However, parameters such as sugar content, polarized sugar yield, white sugar content, and white sugar yield were the highest in condition B2, treated with an agent containing soil bacteria. Furthermore, biostimulants positively affected the leaf area index, with significant growth observed, especially in condition B3. Another important finding was that in the interaction analysis, the biostimulants had positive effects in dry conditions and on elevated values of traits of Alpaca variety caused by treatment in condition B2. In terms of relationships between individual parameters, an interesting finding was that there was only a weak relationship between root yield and sugar content (Rs = 0.0715), which indicates that biostimulants increase production size while maintaining or increasing its quality.

## 1. Introduction

Sugar beet (*Beta vulgaris* subsp. *vulgaris*) is a biennial, commercially important root crop that secures nearly 20% of global sugar production [1]. From a regionalization perspective, it is grown particularly in temperate conditions [2] but also in dry and semi-dry climatic conditions [3]. The importance of this agricultural crop is highlighted especially by its ability to store a large amount of saccharose in its roots [4], which is subsequently used for sugar production [5]. The utility of sugar beet is very wide because its primary use is not as a saccharose source, but it is known for its multiple uses in many areas of manufacturing [6]. FAO [7] states that global sugar beet production achieved nearly 253 million tons.

Drought stress is one of the most limiting factors that cause losses in yield [8] and quality [9] of the sugar beet. By contrast, under the influence of drought stress, an increased accumulation of compatible solutes, such as glucose, fructose, amino acids, potassium, and sodium, can be observed [10]. Moreover, Kenter et al. [11] emphasize that the development of the sugar beet yield largely depends on weather conditions during the entire vegetation period. Another important factor considered to influence sugar beet yield is the variety [12]. Moreover, growing varieties with higher drought tolerance can mitigate the negative impact of adverse weather conditions, although the breeding process is time-consuming and costly [13]. In particular, higher drought tolerance should lead to higher yield stability across years and various environmental conditions [14]. The level of variability of anatomical or morphological traits of the genetic material from the perspective of water usage effectiveness and drought reaction can be used as potential markers for genotype selection of sugar beet with improved tolerance for water scarcity [15].

To increase the usability of nutrients and production and to increase the tolerance for biotic and abiotic stress, biostimulants, extracts with a variety of bioactive substances, can be used in agriculture [16,17]. The application of biostimulating agents in small doses directly on soil or the plant’s leaf area can positively affect plant growth [18]. These products for plant protection against the influence of abiotic stress are available in various forms with various ingredients. Still, they are generally classified into several basic groups: humic and fulvic acids, products containing hormones, amino acid compounds, and agents containing selected microorganisms [19]. The influence of humic and fulvic acids is manifested as better reception of nutrients and water [20,21]. Under stress conditions (high salinity), applying humic acid or seaweed extracts can increase vegetative plant growth and increase the fresh or dry leaf matter and antioxidant activity of enzymes [22]. Biostimulants with specific amino acid compounds affect the plant by modulating the capture and assimilation of nitrogen using regulation enzymes that contribute to nitrogen assimilation and their structural genes and thus affect the signaling pathways of obtaining nitrogen in the roots of sugar beets [23]. Some biostimulants based on amino acid compounds also have chelation effects (e.g., proline), which protect the plant from the negative effects of heavy metals but also contribute to mobility and obtaining micronutrients from soil tissue [19]. Biostimulants with seaweed as their basic ingredient affect the crop by contributing to gel creation, water retention, and soil aeration [24].

Although the global biostimulants market is one of the fastest developing agriculture-related industries, with 2.5 billion USD revenue in 2019, which surpasses the growth speed of the inorganic fertilizer market several times [25], it is important to continue to focus on biostimulants.

The application of biostimulants in agriculture is well known. However, there is still a lot of room for research in the field of combining biostimulants of different origins, and this is the novelty of this study and the obtained results. Therefore, an experiment was conducted applying two different biostimulants on sugar beet vegetation to increase its production and quality. All operations were conducted in field conditions, which raises the credibility of the results and their applicability to practice to ensure the profitability and sustainability of sugar beet production.

## 2. Results

### 2.1. Yield, Quality, and Leaf Area Index of Sugar Beet

As shown in Figure 1, significant differences were found in the results of production parameters of sugar beet treated with biostimulants; this was confirmed through the ANOVA (Table 1). However, in most cases, a positive effect of biostimulants on sugar beet traits was found. In condition B3, the significantly highest two-year average RY was found compared with the control condition. The highest values of SC, PSY, WSC, and WSY were found in condition B2 (Figure 1), all significantly different from those in the control condition. The components for molasses formation are an important criterion for sugar beet quality and contribute to decreasing the sugar yield. From this perspective, an interesting finding is that the application of biostimulants in most cases (except for condition B1) increased the values of sodium, potassium, and α-aminoN in sugar beet pulp. However, this fact had no significant influence on the final production results.

LAI is an important determinant of foliage productivity during vegetation. This study found a significant effect of biostimulants on LAI (Table 1). The observed treatments found the highest average value of LAI during the two-year experiment in condition B3. Furthermore, the post-hoc analysis confirmed the significance of the difference between the LAI value found in the control condition and those in the conditions treated with biostimulants (Figure 2).

### 2.2. Interactions between the Observed Experimental Factors

#### 2.2.1. Biostimulant × Year Interaction

The weather conditions of the year considerably affect field crop production. Therefore, it is essential to investigate the interaction between the biostimulants and experimental years. According to Kožnar and Klabzuba [26], although the year 2018 can be characterized as thermally normal, regarding rainfall, it is considered dry (Table 2). This probably had a decisive effect on the production parameters of the sugar beets in this study. Nevertheless, a significant biostimulant × year interaction on the RY of sugar beets was not found, and in Figure 3a, considerable differences in the results of this parameter can be seen. Higher values of RY were found in all conditions in the year 2019 compared with the year 2018. The absolute highest value of RY was found in the interaction B3 × 2019. An interesting finding was that the interaction biostimulant × year significantly affected all other production parameters except for LAI (Table 1). The highest average value of SC and WSY were found in the B2 × 2018 interaction (Figure 3b,d). The studied parameters of PSY and WSY were the highest, similarly to RY, in the B3 × 2019 interaction (Figure 3c,e). Large differences were found in studying the effects of the biostimulant × year interaction on components of molasses formation (Figure 3f–h). This might have been caused by the biostimulant composition or the speed of the migration of individual components in different weather conditions of the observed year. Nevertheless, it can be stated that the lowest values of sodium, potassium, and α-aminoN were found in the B1 condition, although in different years. 

Expected results were found regarding the effect of the biostimulant × year interaction on LAI because dry weather in 2018 was expressed in a considerable decrease in the leaf area of the sugar beet. The highest value of LAI was measured in the B3 × 2019 interaction (Figure 4). Furthermore, it can be concluded that the LAI values in all conditions in 2019 were higher than those in 2018.

#### 2.2.2. Biostimulant × Variety Interaction

The genetic base of the investigated varieties treated using the biostimulant agent was fully expressed on the monitored parameters of the sugar beet. Whereas quantitative production characteristics, except for WSY, were not significantly affected by this interaction (Table 1), nearly all qualitative indicators were significantly affected. It is clear from the results that significantly higher values of SC, PSY, WSC, and WSY were found in the B2 × Alpaca interaction (Figure 5b–e). The highest RY were recorded in the B3 × Gorila and B3 × Alpaca interactions, with significant differences compared with the interactions of studied varieties in the control condition (Figure 5a). High variability of results was observed depending on the specific interaction on the values of molasses formation components. This was expected considering the genetic characteristics of the varieties included in the experiment. In general, it can be confirmed that the Gorila variety reacted more prominently to the potassium and α-aminoN content when treated with biostimulants (Figure 5g,h), whereas considering sodium content, a considerable (positive) effect was observed in the B1 × Alpaca interaction (Figure 5f).

The biostimulant × variety interaction had no statistically significant effect on LAI (Table 1); however, as can be seen in Figure 6, all conditions treated with biostimulating substances had higher LAI values than those in the control conditions. The highest absolute LAI value was found in the B3 × Alpaca interaction with a significant difference from other combinations, except B3 × Gorila (Figure 6).

#### 2.2.3. Biostimulant × Year × Variety Interaction

The investigation of all three factors on the parameter results of the sugar beet is important from the perspective of recommendations for specific growing conditions. An interesting finding of this study was that the three-way interaction significantly affected the qualitative parameters of the sugar beet (SC, WSC, Na^+^, K^+^, and α-aminoN). In contrast, the effect on quantitative parameters (RY, WSY, PSY) was not found (Table 1). Overall, it can be concluded that the highest values of SC, PSY, WSC, WSY, and Na+ were found in the B2 × 2019 × Alpaca interaction (Figure 7). The highest sugar beet RY was achieved in condition B3, the year 2019, and variety Gorila (Figure 7a).

The LAI results of the biostimulant × year × variety interaction follow the results of the RY parameter. The highest LAI value was found in the B3 × 2019 × Gorila (Figure 8), which confirms the close relationship between these parameters.

### 2.3. Correlation Analysis between Sugar Beet Traits

Based on the results presented in the Spearman correlation matrix (Figure 9), it can be confirmed that there are very strong relationships between RY and WSY (Rs = 0.8046) and RY and PSY (Rs = 0.8346). Furthermore, a very strong correlation was found between SC and WSC (Rs = 0.9718), and a moderate relationship was found between SC and WSY (Rs = 0.5921) and SC and PSY (Rs = 0.5524). An important finding of this multiyear experiment was that the relationship between RY and SC was very weak (Rs = 0.0715). 

As shown in Figure 10, a correlation analysis between RY and LAI and SC and LAI confirmed the results presented above regarding three-way interactions. Whereas a medium-strong correlation was found between RY and LAI (Rs = 0.5604), a weak correlation was found between SC and LAI (Rs = 0.1225).

## 3. Discussion

In general, the growing of field crops is influenced by many biotic and abiotic stresses. However, one of the greatest limits in agriculture is currently the ongoing climate change. In the temperate zone, this phenomenon is accompanied by long periods of drought and higher temperatures during the plant vegetation period, which cause a decrease in their production potential. Therefore, a multiyear experiment was conducted, in which the effect of biostimulant application on the increase in production and quality of the sugar beet was observed. Furthermore, this approach, in combination with suitable genetic material, can be an effective measure in the fight against climate change. 

Many authors state that environmental conditions can be considered the most important limiting factors to the production and quality of field crops [4,27,28,29,30]. The results of this study (Table 5) clearly confirmed that the year’s weather conditions had a significant effect on all sugar beet parameters except sodium content. This fact has also confirmed that it is necessary to investigate the possibility of mitigating adverse weather conditions effects on growing sugar beets and other field crops. 

Much attention has recently been given to biostimulant applications in plant production regarding mitigating the biotic and abiotic stresses [31,32,33,34]. The results of this trend are lowered doses of industrial fertilizers, and the plants are also increasing their tolerance to abiotic and biotic stresses [16]. This study found a significant effect of biostimulants on all monitored sugar beet parameters (Table 5). Furthermore, due to the foliar application of biostimulants, a significant increase in RY (B2 and B3; Figure 2a) and SC (Figure 2b) was observed. This corresponds with the findings of several authors who confirm that biostimulants positively affect the quantity of production and the quality of plants [17,35,36,37]. In contrast, studies by Pulkrábek et al. [38] and Rašovský and Pačuta [39] show a less prominent effect of biostimulants on the quality of sugar beet production. Similar to the parameters mentioned above, significantly higher values of PSY and WSC were found compared with the control conditions (Figure 2c,d). Due to biostimulant application containing soil bacteria (B2), the content of potassium and α-aminoN in beet juice significantly increased compared with the control conditions (Figure 2g,h). Furthermore, Artyszak and Gozdowski [37] state that molasses formation components and SC have a lower influence on the production of the final product than RY. This was also shown in this study, in which the increased values of molasses formation components had no determining effect on the WSY (Figure 2e), and its value was determined rather by RY. However, the finding of Hoffmann [40] that the concentration of molasses formation components decreases with higher root volume and mass was not supported.

In field crop production, LAI, defined as the total leaf area per unit of ground area, is often used to quantify the vegetation structure [41]. Furthermore, as indicated by Sharma et al. [42], an often observed effect of biostimulant application is the growth of the leaf area and the chlorophyll content in the leaves. In this study, a significant effect of biostimulants on LAI was confirmed (Table 5), although a significant increase compared with the control condition was found only in the condition with a biostimulant combination (Figure 3). The correlation analysis highlights the connection between the highest LAI value and RY of sugar beet in condition B3, in which a medium-strong relationship was found between these two parameters (Figure 10). The positive effects of biostimulant application on LAI in the system of growing field crops were confirmed by several studies [43,44,45].

Interesting findings were observed in the monitored factor interactions. Although there is an insufficient amount of information on the effect of biostimulant × year on production parameters of sugar beets, this study shows that biostimulation protection positively affected most parameters even in a dry year (Figure 4). Similarly, as in the previous case, there is little research on the effect of biostimulant × variety interaction. However, from the perspective of production quality, in particular, it is clear that a specific agent with a selected variety may significantly increase the quality of the sugar beet (Figure 6). 

More research focus has been given to determining the effect of the interaction of genotype and environment. Curcic et al. [46] state that the plant growth, development, and ultimately yield of sugar beets result from the genetic composition, environmental effects, and interactions. Furthermore, the singularity of the genotype in interaction with the environment is always present in crop production, which causes that different genotypes to achieve different results in different environmental conditions [47]. Thus, this interaction is considered an important part of the process of breeding plants [48]. However, Hoffmann et al. [12] report that the yield and quality of the sugar beet are primarily dependent on the environment and variety. Their relationship did not have a considerable effect on the resulting parameters; therefore, it seems that there is no specific suitability of sugar beet genotypes for specific environmental stress conditions. In this study, the influence of the environment and genotype on the production results was confronted with biostimulation protection (Figure 8 and Figure 9). The results clearly show that the highest values of most parameters were found in condition B2 in 2019 with the Alpaca variety. However, Studnicki et al. [49] concluded that it is not possible to list one cultivar characterized by wide adaptation to all environmental conditions with a relatively high and stable WSY. Concurrently, the cultivars that are considered the most stable are those characterized by negligible interaction of the environment and genotype [50].

The negative relationship between the quantity and quality of production is a well-known fact in agricultural production [51,52]. Regarding this perspective, different results were found in this experiment. A weak relationship was found in terms of the most important production parameters of the sugar beet, RY and SC (Figure 10). This requires further attention as it suggests that using biostimulants used in this experiment can lead to not only a production increase but also an increase in its quality.

## 4. Materials and Methods

### 4.1. Experimental Sites

The field experiment was conducted in 2018 and 2019 at a research station SUA in Nitra (E 18°09′, N 48°19′) (Figure 11). This is a location in a maize-producing area and part of the Danubian Lowland. The area is characterized by a dry and warm climate, especially during the vegetation period. Rainfall and average temperatures during vegetation periods of the experimental years were recorded using a hydro-meteorological station directly at the experimental base and are listed in Table 2. Lower quality medium-heavy loamy soil with low humus content and weakly acidic pH can be found on the property.

### 4.2. Agrotechnical Traits and Soil Analysis

After the harvest of the pre-crop (*Triticum aestivum* L.) in the autumn of each year, the stubble was plowed under. Subsequently, mid-depth tillage was performed with the concurrent placement of phosphorus and potassium fertilizers and stable manure (50 t ha^−1^). Nitrogen fertilization was applied during the pre-sowing preparation of the soil in the spring. The nitrogen doses were calculated using the nitrogen balance method based on five random samples taken across the plot and their analysis (Table 3) for an expected root yield of 70 t ha^−1^. The ammonia form of nitrogen was determined calorimetrically using Nessler’s reagent [53], and the nitrate form calorimetrically using phenol 2.4-disulfonic acid [54]. The Mehlich III test was used to determine the phosphorus and potassium content in the soil [55]. Furthermore, soil samples were collected each year to determine the soil reaction using a 1-molar solution of potassium chloride [56] and the humus content based on the Tjurin method [57]. Before planting the sugar beet, shallow aerating of the soil was performed. All agrotechnical operations during vegetation (e.g., application of herbicides and fungicides) were conducted according to the requirements and needs of the sugar beet vegetation.

### 4.3. Genetic Materials and Experimental Design

Varieties of Alpaca and Gorila (SESVanderHave International BV) were selected for the experiment. Alpaca is a variety of sugary types, resistant to Cercospora leaf spot and tolerant to abiotic stress. Gorila is a variety of transitional types, highly resistant to droughts and infestation of the pest *Tetranychus urticae*. The experiment was conducted using the randomized complete block design [58] in three repetitions using a 12-row drill with seed spacing in a row of 0.18 m and row spacing of 0.45 m. Each experimental lot had an area of 32.4 m^2^ (6 m length × 5.4 m width). The dates of sowing were selected based on the conditions at the base as 18 April 2018 and 2 April 2019.

### 4.4. Biostimulant Characteristics

The effects of biostimulants of various origins were investigated in the experiment. The individual types were labeled B0-B3. B0 was the control condition without biostimulant application. B1 was the condition with biostimulant with high content of organic oligopeptides (100 g L^−1^), a mixture of essential amino acids, tryptophan, lysine and threonine (20 g L^−1^), and potassium as K_2_O (8.35% in dry matter). Condition B2 was treated with a biostimulant characterized by a unique combination of amino acids, alanine, tryptophan, and arginine (50 g L^−1^). Further, it contained an extract of soil bacteria PHB (poly-beta-hydroxybutyrate) (6 g L^−1^). B3 was the condition in which the effect of the combination of the treatments used in B1 and B2 on the sugar beet parameters was investigated. Biostimulants were applied foliarly twice during the vegetation period in selected growth phases (Table 4).

### 4.5. Leaf Area Index (LAI) Estimation

The leaf area index (LAI) was measured in five different growth phases (Table 5) to evaluate the overall and seasonal productivity. The LAI is characterized as the total one-sided leaf tissue area per unit of soil surface [59]. The device SS1 SunScan Canopy Analysis System (Delta-T Devices Ltd., Cambridge, United Kingdom) was used to measure LAI like Ariza-Carricondo et al. [60].

### 4.6. Harvesting and Production Analysis

Plants were harvested manually; two representative rows were plowed out at each plot. The root yield (RY) was determined directly at the plot and calculated per hectare. Samples from each plot were analyzed for quality (sugar content (SC), α-aminoN, K^+^, Na^+^) on the Venema Analyzer IIIG (Venema Consulting, Groningen, Netherlands) like Barlog et al. [61].

Based on the obtained values, further production parameters were calculated. White sugar content (WSC) was calculated based on the following formula [62]:WSC = SC − [(K + Na) × 0.343 + (0.094 × α-aminoN) + 0.29] (%) (1)

The polarized sugar yield (PSY) and the white sugar yield (WSY) were calculated according to Bajči et al. [63]:PSY = 0.01 × (RY × SC) (t ha^−1^)(2)
WSY = 0.01 × (RY × WSC) (t ha^−1^)(3)

### 4.7. Statistical Analysis

The experiment results were compiled and analyzed in the statistical program Statistica 10 (StatSoft, Inc., Tulsa, OK, USA). Multifactor analysis of variance (ANOVA) was used to determine the effect of the main factors of the experiment on the monitored sugar beet parameters. To determine any significant within factor differences, post-hoc analysis using the LSD Tukey test with significance level α = 0.05 were performed. Multiple regression analysis and nonparametric correlation analysis (Spearman coefficient) were performed to determine the factor relationships between selected sugar beet parameters.

## 5. Conclusions

The results of this experiment substantiate biostimulant application in growing sugar beets not only from a production perspective but also from a sustainability perspective. Humic acid-based stimulants with an increased proportion of amino acids or with soil bacteria content benefited the production quantity, quality, and LAI. A positive finding was that an increase in sugar beet traits was observed even in a year with adverse weather conditions. Regarding white sugar production, the highest values were observed in condition B2. Furthermore, this effect was even more prominent in combination with the conditions in 2019 and with the Alpaca variety. The observation of relationships between sugar beet traits did not support the accepted negative correlation between RY and SC. This suggests that biostimulants increase the production quantity while maintaining or increasing quality. In connection with increased pressure to decrease the use of industrial fertilizers, the need for biostimulant application continues to increase.

## Figures and Tables

**Figure 1 plants-11-02222-f001:**
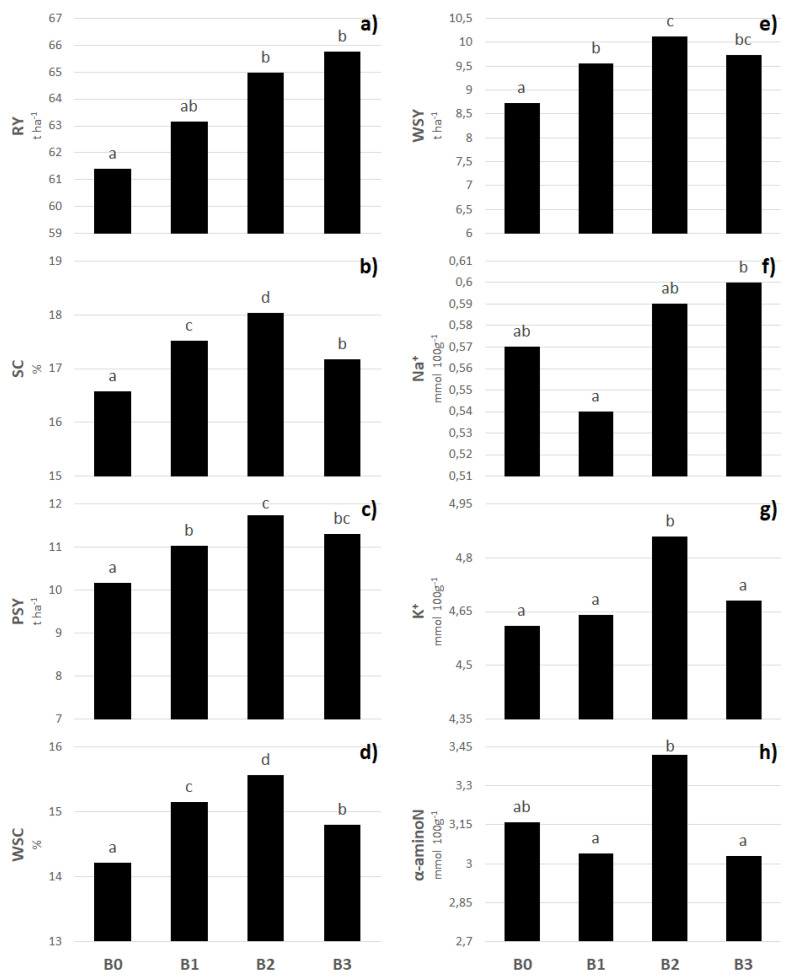
Different sources of biostimulants influence on production traits of sugar beet. (RY represented root yield (**a**); SC—sugar content (**b**); PSY—polarized sugar yield (**c**); WSC—white sugar content (**d**); WSY—white sugar yield (**e**); Na^+^—sodium content (**f**); K^+^—potassium content (**g**); α-aminoN—α-aminonitrogen content (**h**); B0—control; B1—humic acids, essential amino acids; B2—essential amino acids biopolymers, soil bacteria; B3—humic acids, essential amino acids biopolymers, soil bacteria). The figure was obtained at a level of significance α = 0.05.

**Figure 2 plants-11-02222-f002:**
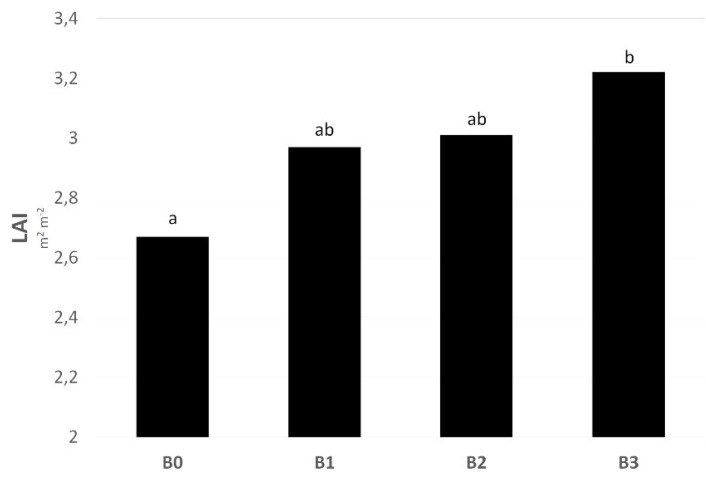
Different sources of biostimulants influence on sugar beet leaf area index (LAI). B0—control; B1—humic acids, essential amino acids; B2—essential amino acids biopolymers, soil bacteria; B3—humic acids, essential amino acids biopolymers, soil bacteria. The figure was obtained at a level of significance α = 0.05. Different letters above the columns indicate the significance of the difference.

**Figure 3 plants-11-02222-f003:**
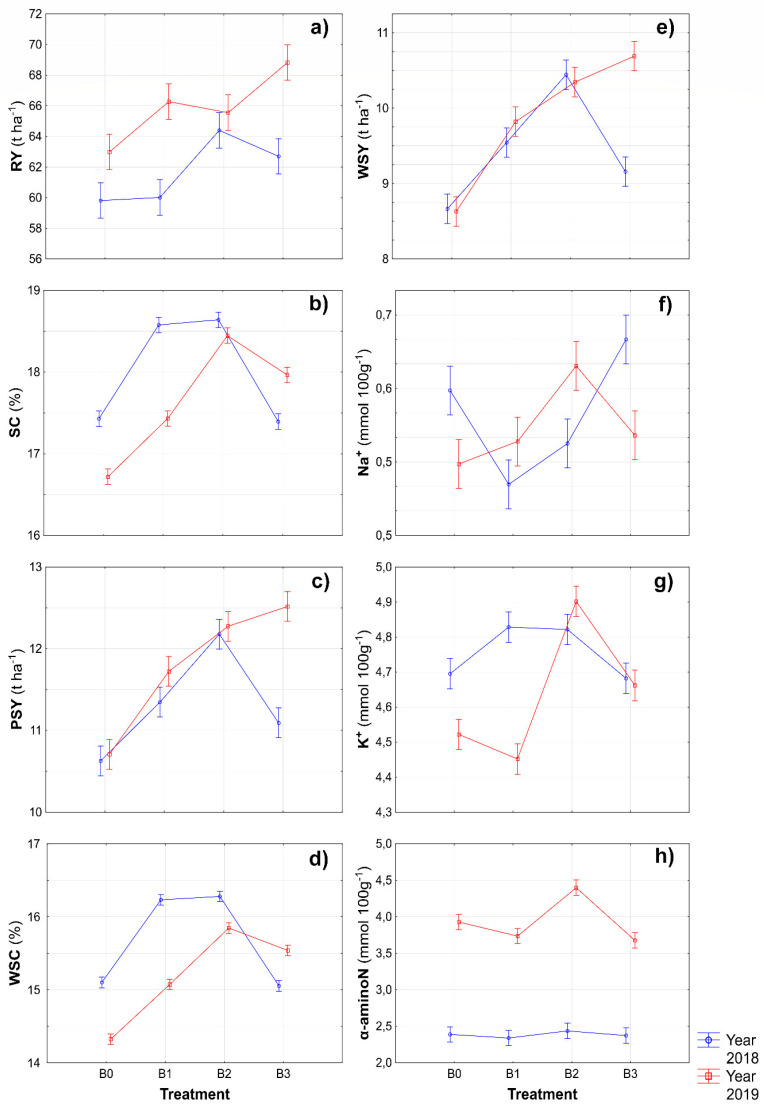
Interactions between biostimulants and experimental years for root yield (**a**), sugar content (**b**), polarized sugar yield (**c**), white sugar content (**d**), white sugar yield (**e**), sodium content (**f**), potassium content (**g**) and α-aminonitrogen content (**h**). B0—control; B1—humic acids, essential amino acids; B2—essential amino acids biopolymers, soil bacteria; B3—humic acids, essential amino acids biopolymers, soil bacteria Figure was obtained at a level of significance α = 0.05.

**Figure 4 plants-11-02222-f004:**
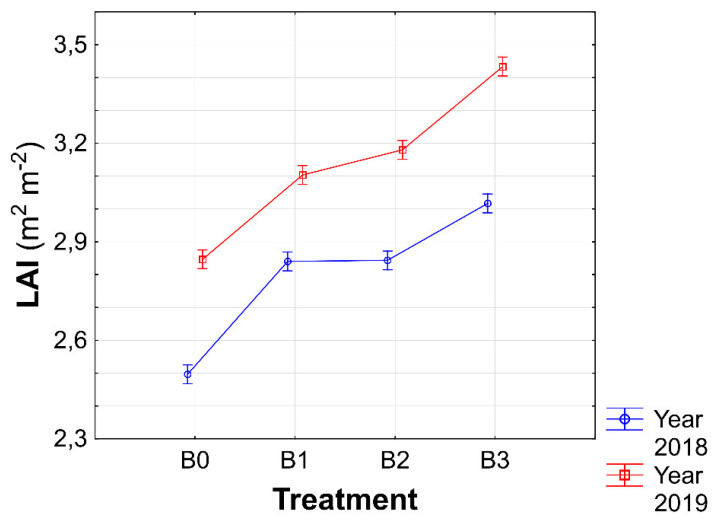
Interactions between biostimulants and experimental years for leaf area index (LAI). B0—control; B1—humic acids, essential amino acids; B2—essential amino acids biopolymers, soil bacteria; B3—humic acids, essential amino acids biopolymers, soil bacteria Figure was obtained at a level of significance α = 0.05.

**Figure 5 plants-11-02222-f005:**
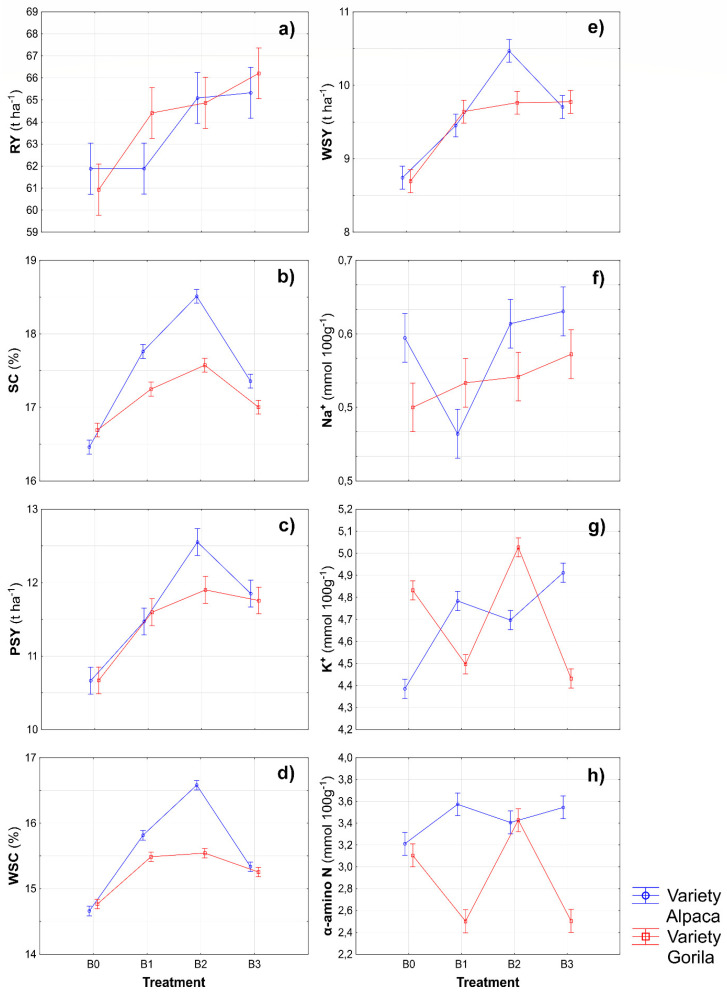
Interactions between biostimulants and varieties for root yield (**a**), sugar content (**b**), polarized sugar yield (**c**), white sugar content (**d**), white sugar yield (**e**), sodium content (**f**), potassium content (**g**) and α-aminonitrogen content (**h**). B0—control; B1—humic acids, essential amino acids; B2—essential amino acids biopolymers, soil bacteria; B3—humic acids, essential amino acids biopolymers, soil bacteria. The figure was obtained at a level of significance α = 0.05.

**Figure 6 plants-11-02222-f006:**
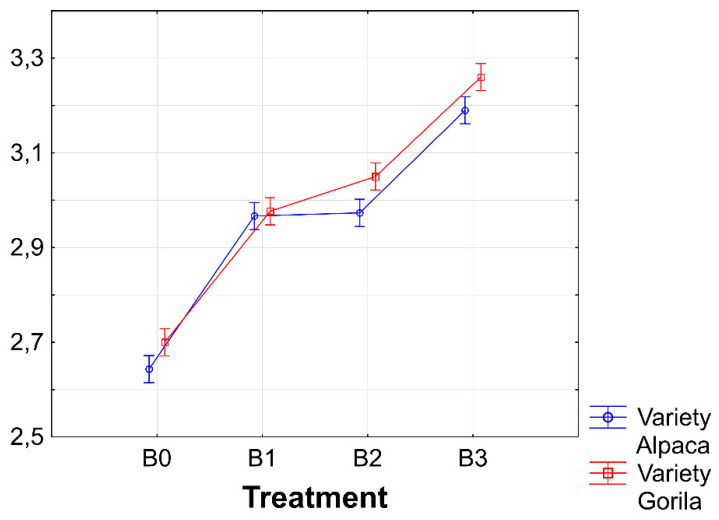
Interactions between biostimulants and varieties for leaf area index (LAI). B0—control; B1—humic acids, essential amino acids; B2—essential amino acids biopolymers, soil bacteria; B3—humic acids, essential amino acids biopolymers, soil bacteria. The figure was obtained at a level of significance α = 0.05.

**Figure 7 plants-11-02222-f007:**
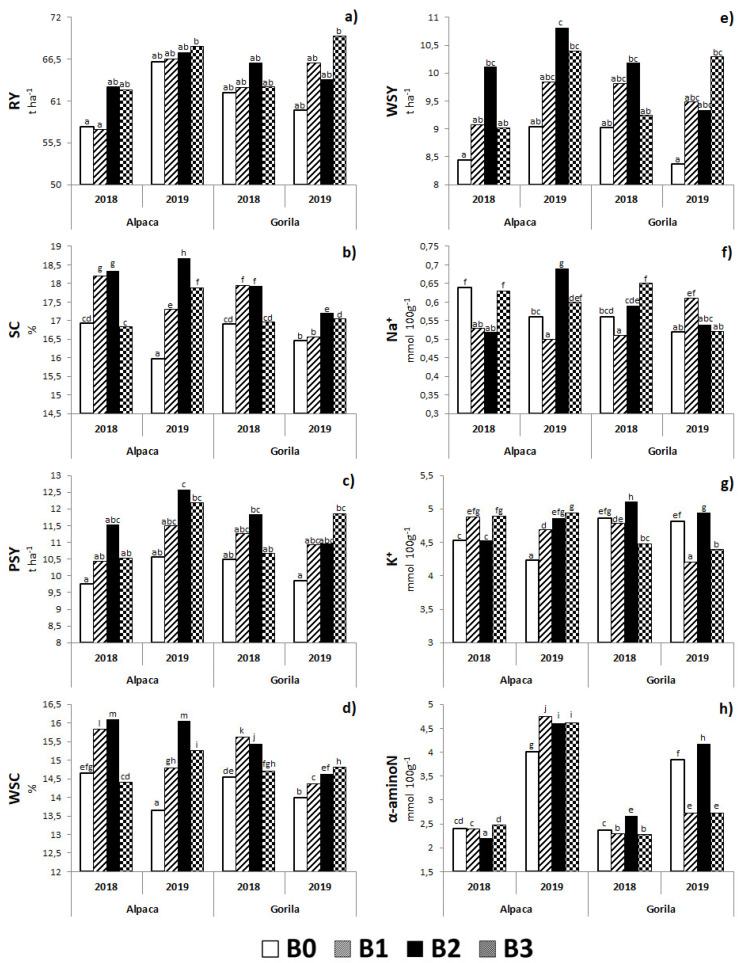
Interactions between biostimulants, years, and varieties for root yield (**a**), sugar content (**b**), polarized sugar yield (**c**), white sugar content (**d**), white sugar yield (**e**), sodium content (**f**), potassium content (**g**) and α-aminonitrogen content (**h**). B0—control; B1—humic acids, essential amino acids; B2—essential amino acids biopolymers, soil bacteria; B3—humic acids, essential amino acids biopolymers, soil bacteria. The figure was obtained at a level of significance α = 0.05.

**Figure 8 plants-11-02222-f008:**
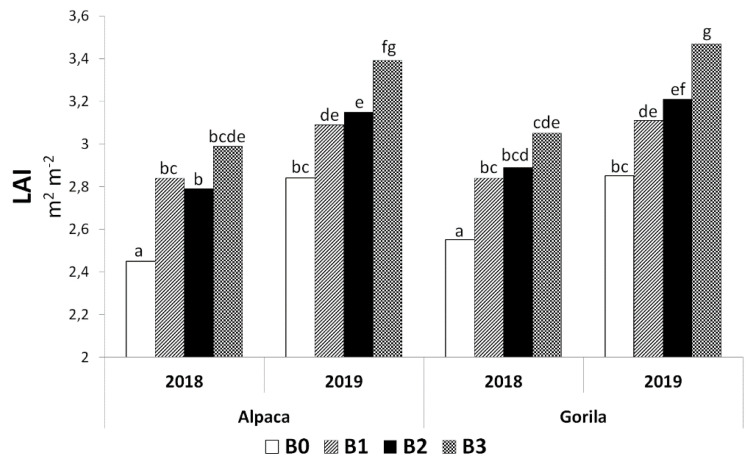
Interactions between biostimulants, years, and varieties for leaf area index (LAI). B0—control; B1—humic acids, essential amino acids; B2—essential amino acids biopolymers, soil bacteria; B3—humic acids, essential amino acids biopolymers, soil bacteria. The figure was obtained at a level of significance α = 0.05.

**Figure 9 plants-11-02222-f009:**
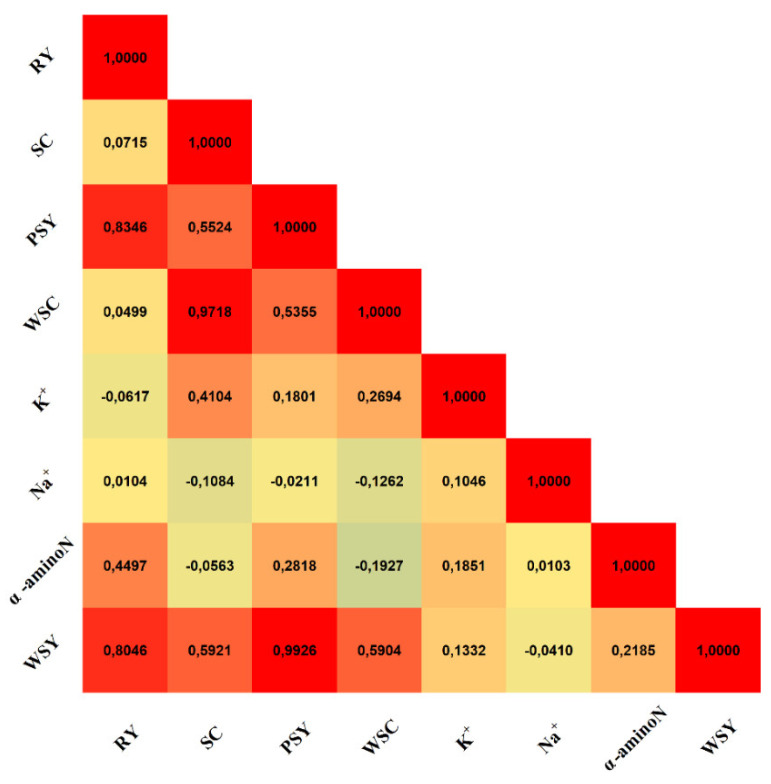
Relationships among sugar beet production traits.

**Figure 10 plants-11-02222-f010:**
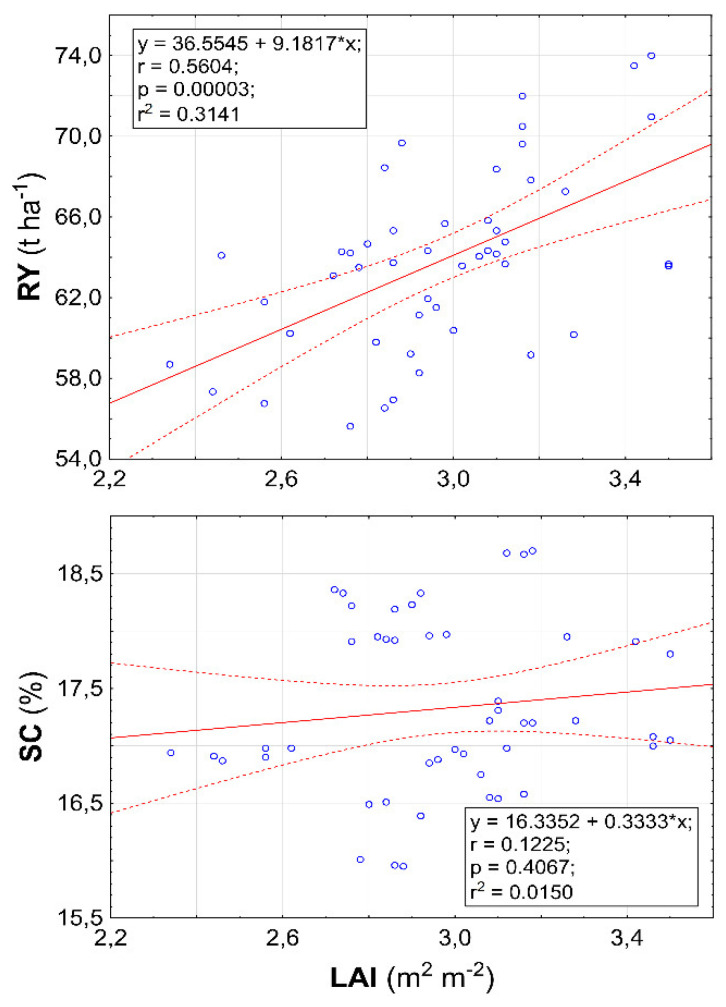
Relationships between leaf area index (LAI) and root yield (RY), and sugar content (SC), respectively. Solid lines represent the linear relationship between parameters. Linear equations, correlation coefficient, probability, and regression are inserted inside the figures.

**Figure 11 plants-11-02222-f011:**
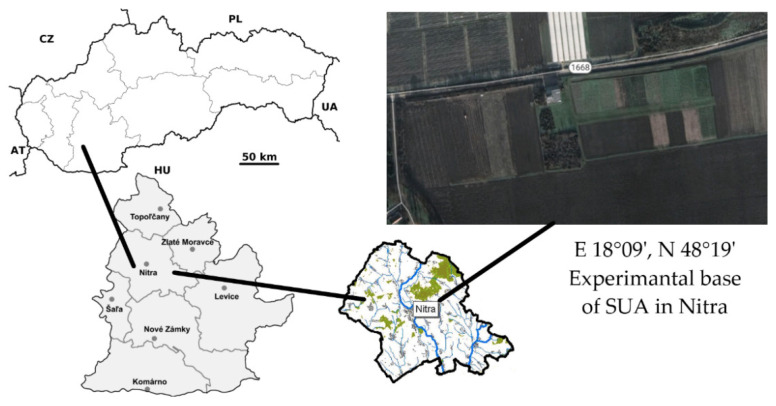
Location of experimental station of the Slovak University of Agriculture in Nitra.

**Table 1 plants-11-02222-t001:** Analysis of variance (ANOVA) in sugar beet experiment.

Source of Variation	Parameters								
	RY (t ha^−1^)	SC (%)	PSY (t ha^−1^)	WSC (%)	WSY (t ha^−1^)	K^+^ (mmol 100 g^−1^)	Na^+^ (mmol 100 g^−1^)	α-aminoN (mmol 100 g^−1^)	LAI (m^2^ m^−2^)
	*p*-Values								
B	0.0046	0.0000	0.0000	0.0000	0.0000	0.0000	0.0294	0.0034	0.0024
Y	0.0000	0.0000	0.0008	0.0000	0.0058	0.0006	0.4845	0.0000	0.0010
V	0.5004	0.0000	0.2462	0.0000	0.2741	0.9359	0.1109	0.0000	0.6025
*B × Y*	0.1071	0.0000	0.0000	0.0000	0.0010	0.0002	0.0032	0.0225	0.9630
*B × V*	0.4778	0.0000	0.1801	0.0000	0.0386	0.0000	0.0874	0.0000	0.9957
*Y × V*	0.0022	0.0010	0.0000	0.0052	0.0001	0.0048	0.2262	0.0000	0.9093
*B × Y × V*	0.2272	0.0000	0.5231	0.0000	0.5195	0.0000	0.0000	0.0000	0.7676

RY—root yield; SC—sugar content; PSY—polarized sugar yield; WSC—white sugar content; WSY—white sugar yield; Na^+^—sodium content; K^+^—potassium content; α-aminoN—α-aminonitrogen content. B—biostimulant; Y—year; V—variety. Analysis was obtained at a level of significance α = 0.05.

**Table 2 plants-11-02222-t002:** Precipitation and temperature during the sugar beet vegetation.

Year	April	May	June	July	August	September	October	
	Precipitation (mm)							Sum
2018	12.2	14.6	97.5	12.9	3.0	57.2	14.4	211.8
2019	21.4	134.8	29.0	52.2	64.0	52.8	17.8	372.0
Normal *	41.6	56.0	66.2	59.3	54.2	43.1	41.0	361.4
	Temperature (°C)							Mean
2018	13.2	15.9	17.8	18.3	19.0	13.8	9.7	15.4
2019	9.4	9.3	18.7	18.0	18.4	12.6	8.7	13.6
Normal *	10.4	15.2	18.3	20.0	19.7	15.5	10.2	15.6

* Normal–representing climatic normal of Dolná Malanta location from 1951–2000.

**Table 3 plants-11-02222-t003:** Soil properties in experimental field.

Year	Macronutrient Content mg kg^−1^			pH	Humus Content (%)
	N_total_*	P	K		
2017/2018	25.18	93	385	6.28	1.72
2018/2019	10.00	63	315	6.69	1.60

N_total_*—is represented as the sum of the nitrate and ammoniac forms of nitrogen.

**Table 4 plants-11-02222-t004:** Biostimulants treatment application.

Treatment	Source of Biostimulants	Application Doses (L ha^−1^)	Form of Application	Growth Phase of Sugar Beet
B0	-	**	foliar	BBCH 19 and 33
B1	humic acids, essential amino acids	10 *		
B2	essential amino acids biopolymers, soil bacteria	0.5 **		
B3	combination B1 and B2	10 * + 0.5 **		

* dose of 250 L H_2_O ha^−1^, ** dose of 200 L H_2_O ha^−1^.

**Table 5 plants-11-02222-t005:** Growth phases of leaf area index measurement.

Measurement	Growth Phase	Definition
1.	BBCH 19	Nine and more leaves unfolded
2.	BBCH 31	Beginning of canopy cover: leaves cover 10% of ground
3.	BBCH 33	Leaves cover 30% of ground
4.	BBCH 39	Canopy cover complete
5.	BBCH 47	Sugar beet has reached technological maturity

## Data Availability

The data presented in this study are available upon request from the corresponding author.

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
