# Peer review of "Quantity and Quality Changes in Sugar Beet (Beta vulgaris Provar. Altissima Doel) Induced by Different Sources of Biostimulants"

_plants, 2022, doi:10.3390/plants11172222_

Round 1

Reviewer 1 Report

The paper entitled “Quantity and Quality changes in Sugar Beet (Beta vulgaris provar. Altissima Doel) Induced by Different Sources of Biostimulants” is focused on investigation of the effect of foliar treatments with different types of biostimulants on the growth and quality of sugar beet. Authors present interesting studies that can be used for effective sustainable production of sugar beet under adverse weather conditions and to improve the quality of sugar beet.

The manuscript is well prepared, nicely organized and written. The paper needs however a minor revision. Please reexamine the paper taking into account the following:

1. Abstract, L14: I recommend to add he constituents of biostimulant in B3 treatment. As example, “B3 (combination of humic acids, essential amino acids, biopolymers, soil bacteria).

2. Introduction: Please mention in the last paragraph of Introduction the novelty of presented study.

3. Legend to the Figure 1: Please add the information about the treatments B0, B1, B2, and B3.

4. L 298: “in 2018 a 2019” please correct to “in 2018 and 2019”.

5. How many samples you took for analysis?

6. Table 4: Please add the space in “aminoacids”.

7. Subsection 4.7: Please add the information about the number of replicates in experiment.

8. I think they are way to many references for a paper like this, they should be reduce up to 50 if its possible with in the last 10 years.

Author Response

Dear reviewer,

thank you for the detailed analysis of our manuscript and the comments that helped improve its quality.

  1. Abstract, L14: I recommend to add he constituents of biostimulant in B3 treatment. As example, “B3 (combination of humic acids, essential amino acids, biopolymers, soil bacteria).

Fully agree with the comment. We edited the sentence based on your recommendation.

  1. Introduction: Please mention in the last paragraph of Introduction the novelty of presented study.

Thank you for your comment, the novelty of our research is stated in the last paragraph of the Introduction.

  1. Legend to the Figure 1: Please add the information about the treatments B0, B1, B2, and B3.

Thank you for your comment, information about individual treatments has been added to each figure.

  1. L 298: “in 2018 a 2019” please correct to “in 2018 and 2019”.

Thanks for the notification, the error has been fixed.

  1. How many samples you took for analysis?

Information on the number of samples was added to MM.

  1. Table 4: Please add the space in “aminoacids”.

Thanks for the notification, the error has been fixed.

  1. Subsection 4.7: Please add the information about the number of replicates in experiment.

Thanks for the comment. The design of the experiment and the number of repetitions is mentioned in chapter 4.3

  1. I think they are way to many references for a paper like this, they should be reduce up to 50 if its possible with in the last 10 years.

Thank you for your comment, we believe that all references in the manuscript have their meaning and place.

Reviewer 2 Report

Dear authors, I have read with interest your manuscript and I think the topic and the study are particularly relevant. In my opinion the paper is well written, but I have a few questions about the analyses and the graphs you presented.

In Table 1 you are presenting the ANOVA results considering the effect of the singular factors and the interactions. Then you show the same parameters in figure 1, 3, 5 and 7 (and the others for LAI) discussing the different interactions. If there is a significant interaction between all the three factors considered (Time, Variety, Biostimulant) as it is for SC, WSC, K Na and a aminoN it means that I have to look at the graphs in figure 7 and not the other representing the average of two of the factors.

Again, in figure 3, 4, 5, 6 you used the lines, but it might be a little bit confusing because it seems like a trend in time, while in x axes you are reporting the different biostimulants. In my opinion bar charts are a better choice in these cases.

When you insert the letters in the graphs, you can decide if the “a” is the lowest, the “highest”, or the “control”…according to the importance of the representation. Please choose a rule and stick to it because sometimes the “a” is on the highest (figure 1a), sometimes on the lowest (figure 1b) or in the middle (figure 1c,e)…. Please check all the figures.

I think it is better to add in each figures caption what B0-B3 treatments are and what are you representing…the mean of which factors with the number of replicates (n=…). For example, in figure 1 you are representing the mean of varieties and years. But if there is a significant interaction these graphs are not showing something really interesting.

Line 50: biostimulant products are obtained not only from plants extract but there are different raw materials with different origin.

Line 318: please check the punctuation “Mehlich III.”

Line 329: please correct “varieties Alpaca AND Gorila”

Line 359: Plants were harvested, not the experiment…please find another term.

Author Response

Dear reviewer,

thank you for the detailed analysis of our manuscript and the comments that helped improve its quality.

  1. In Table 1 you are presenting the ANOVA results considering the effect of the singular factors and the interactions. Then you show the same parameters in figure 1, 3, 5 and 7 (and the others for LAI) discussing the different interactions. If there is a significant interaction between all the three factors considered (Time, Variety, Biostimulant) as it is for SC, WSC, K Na and a aminoN it means that I have to look at the graphs in figure 7 and not the other representing the average of two of the factors.

Thanks for the comment. Table 1 serves as a companion for the results of the entire research. It shows the overall influence of factors and interactions on sugar beet traits. The figure shows the differences between individual observations. We can agree with your observation that the significance of the 3-interaction is of great importance. However, the presentation of two factors is of great importance especially for practice. If, for example, we exclude the influence of weather conditions (Figure 5), this is important information for someone who grows sugar beet in different conditions.

For this reason, we would like to maintain the current presentation.

  1. Again, in figure 3, 4, 5, 6 you used the lines, but it might be a little bit confusing because it seems like a trend in time, while in x axes you are reporting the different biostimulants. In my opinion bar charts are a better choice in these cases.

Thanks for the comment. Our choice of graphs was that we wanted to emphasize the significance of the differences. This is possible using line segments in this type of graph. If possible, we would like to keep the original figures.

  1. When you insert the letters in the graphs, you can decide if the “a” is the lowest, the “highest”, or the “control”…according to the importance of the representation. Please choose a rule and stick to it because sometimes the “a” is on the highest (figure 1a), sometimes on the lowest (figure 1b) or in the middle (figure 1c,e)…. Please check all the figures.

Fully agree with the comment. Figures have been reviewed and modified based on comments. We thank you.

  1. I think it is better to add in each figures caption what B0-B3 treatments are and what are you representing…the mean of which factors with the number of replicates (n=…). For example, in figure 1 you are representing the mean of varieties and years. But if there is a significant interaction these graphs are not showing something really interesting.

Thank you very much for your comment and notice. Some information has been added based on your comment.

In Figure 1, there is presented the effect of the monitored treatments with biostimulants on the characteristics of sugar beet excluding the influence of other factors. Since this is the main topic of the manuscript, we believe that the figure shows important information.

  1. Line 50: biostimulant products are obtained not only from plants extract but there are different raw materials with different origin.

Fully agree with the comment. The sentence has been corrected.

  1. Line 318: please check the punctuation “Mehlich III.”

Thanks for the heads up. The error has been removed.

  1. Line 329: please correct “varieties Alpaca AND Gorila”

Thanks for the heads up. The error has been removed.

  1. Line 359: Plants were harvested, not the experiment…please find another term.

Fully agree with the comment. The sentence was corrected by request.

Reviewer 3 Report

The manuscript describes the impact of different biostimulant combinations (B0-B4) on sugar beet yield, including the yield of white sugar. The authors also characterized the effect of biostimulants on other morphological and physiological parameters. They have found that combination B3 has the highest yield promotion activity while B2 was best in maximizing sugar yield. Overall the paper is well written, and the authors have chosen appropriate methods and designed experiments very well, in a way that facilitates good statistical analysis of all investigated parameters. However, the author did not mention the exact composition of investigated preparations. Therefore, the composition and the concentration of all compounds are lacking. Thus, the meaning of all results and the discussion with other authors are meaningless, no one can reproduce experiments described by authors, so verification of their discoveries is impossible. Without providing this information, the paper should not be published. The authors may patent the preparation before publishing, if necessary.

Author Response

Dear reviewer,

thank you for the analysis of our manuscript and the comments that helped improve its quality.

„The manuscript describes the impact of different biostimulant combinations (B0-B4) on sugar beet yield, including the yield of white sugar. The authors also characterized the effect of biostimulants on other morphological and physiological parameters. They have found that combination B3 has the highest yield promotion activity while B2 was best in maximizing sugar yield. Overall the paper is well written, and the authors have chosen appropriate methods and designed experiments very well, in a way that facilitates good statistical analysis of all investigated parameters. However, the author did not mention the exact composition of investigated preparations. Therefore, the composition and the concentration of all compounds are lacking. Thus, the meaning of all results and the discussion with other authors are meaningless, no one can reproduce experiments described by authors, so verification of their discoveries is impossible. Without providing this information, the paper should not be published. The authors may patent the preparation before publishing, if necessary.“

Thanks for the comment. Based on your comment, some information regarding the composition of the used preparations was added (chapter 4.4).

Round 2

Reviewer 3 Report

I would like to thank the authors for providing more details on the composition of the biopreparates. However, I do not feel they thoroughly answered the comment.

The authors say:

"B1 was the condition with biostimulant with high content of organic oligopeptides (100 g l-1), essential amino acids (20 g l-1), and potassium.

Condition B2 was treated with biostimulant characterized by a unique combination of essential amino acids (50 g l-1) and biopolymers. Further, it contained an extract of soil bacteria (6 g l-1)."

I would like to know:

For B1, what kind of oligopeptide B1 included, how were they prepared, and from what material or maybe they were synthesized or purchased? What was the potassium concentration in the preparation?

For B2, what kind and concentration of biopolymers were there, how were they prepared or purchased, and from which company? Concerning soil bacteria, what species were used, or how were the cultures prepared?

Without detailed information about the composition of these two biopreparations, no scientific team can reproduce the data included in the paper, so in my opinion, this manuscript still is not ready for publication.

Author Response

Dear Reviewer,

based on your comment, we revised the part of the manuscript with information on the composition of the preparations.

However, we must state that some information is part of the production secret and therefore cannot be published.

We also hope that, in relation to the subject of the manuscript, current information about preparations for further reproduction and comparison is sufficient.

Thank you for your help in improving the quality of this manuscript.